# Allergy to Fungi in Veterinary Medicine: *Alternaria*, Dermatophytes and *Malassezia* Pay the Bill!

**DOI:** 10.3390/jof8030235

**Published:** 2022-02-27

**Authors:** Luís Miguel Lourenço Martins

**Affiliations:** Department of Veterinary Medicine, School of Science and Technology, MED—Instituto Mediterrâneo para a Agricultura, Ambiente e Desenvolvimento, University of Évora, 7000-809 Évora, Portugal; lmlm@uevora.pt

**Keywords:** allergy, *Alternaria*, *Aspergillus*, dermatophytes, fungal allergens, immunocompetence, indoor/outdoor allergens, *Malassezia*

## Abstract

The fungal kingdom comprises ubiquitous forms of life with 1.5 billion years, mostly phytopathogenic and commensals for humans and animals. However, in the presence of immune disorders, fungi may cause disease by intoxicating, infecting or sensitizing with allergy. Species from the genera *Alternaria*, *Aspergillus* and *Malassezia*, as well as dermatophytes from the genera *Microsporum*, *Trichophyton* and *Epidermophyton*, are the most commonly implicated in veterinary medicine. *Alternaria* and *Malassezia* stand as the most commonly associated with either allergy or infection in animals, immediately followed by *Aspergillus*, while dermatophytes are usually associated with the ringworm skin infection. By aiming at the relevance of fungi in veterinary allergy it was concluded that further research is still needed, especially in the veterinary field.

## 1. Introduction

Fungi are living forms and have evolved for approximately 1.5 billion years [1]. Fossil evidence of fungi is scarce, probably because of their easily disrupted soft body nature, frequent microscopic dimension and morphology difficult to distinguish from those of other microbes [2]. The majority of fungal organisms are saprophytic, lacking pathogenicity to plants, humans or animals. However, a small proportion of species may become pathogenic, affecting plants, humans and animals by producing toxins, infecting or causing allergy in humans and animals. Between the genera Alternaria, Mucor, Aspergillus and Fusarium [3], as well as Trichophyton and Microsporum [4], may be found the most frequently involved pathogenic fungal species to plants, humans and animals. Several of the species comprised in those genera are able to cause considerable economic losses to agriculture, with relevant loss of food for consumption, and serious diseases in humans and animals, especially in immunocompromised individuals [3], a Species from the fungal kingdom can be found almost everywhere. Fungal species evolved side by side with other live beings as decomposers of organic matter. By secreting enzymes into their environment, fungal species can extract the available nutrients, mostly carbohydrate metabolites, from other organisms as they are heterotrophic. Other nutrients, such as proteins and lipids, are also digested for subsequent fungal absorption, and in the end, the environment becomes full of the leftovers from fungal digestive proteins. Successively, fungal environmental spread happens through airborne dissemination of spores, hyphae and hyphal fragments, as well as those leftover, reaching almost all places on earth [5]. 

This review aimed at the relevance of fungi in veterinary allergy, where the current available information is really scarce and needs further studies in order to allow a better diagnosis and treatment approach.

## 2. Most Relevant Fungi in Health

The *Aspergillus genus* comprises several of the most common fungal species (e.g., *Aspergillus fumigatus, Aspergillus flavus*, *Aspergillus niger*, *Aspergillus nidulans* and *Aspergillus terreus*) involved in respiratory infection, most often in birds, and it may cause large economic losses in the poultry industry [6]. A recent study by Cheng et al. (2020) [7] highlighted the role of Toll-like receptors (TLR) in the mediated innate immune response associated with *A. fumigatus* infection in chickens, triggering a massive production of several pro-inflammatory cytokines, which leads to severe airsaculitis and infiltrative and granulomatous pneumonia. Species from the genus *Aspergillus* are also involved in respiratory allergies, such as allergic bronchopulmonary aspergillosis, allergic *Aspergillus* sinusitis, IgE-mediated asthma or hypersensitivity pneumonitis [8,9]. These situations may derive from primary sensitization to *Aspergillus* airborne compounds, either indoors or outdoors [3].

The Alternaria genus comprises several phytopathogenic species, affecting the quality of grains, as well as different vegetables, such as tomatoes and peppers, and consequently their economic value [3]. Alternaria species also produce several types of cytotoxic and teratogenic mycotoxins, known to block the synthesis of sphingolipid, by inhibiting the rate-limiting enzyme, ceramide synthase [10], which may also compromise the integrity of the skin barrier [11]. Despite less frequently than Aspergillus, Alternaria is also known for its ability to cause onychomycosis, even in healthy individuals [12], but worse conditions may occur in immunocompromised individuals, where skin infections [12,13], keratomycosis [14] or sinonasal infections may be observed [15]. 

Sensitization to *Alternaria* fungus is also common, with species from this genus as the most frequently associated with type I hypersensitivity, which has been related to exposure in the indoor and outdoor environment, mostly in warm climates [16,17].

Species from the *Fusarium* genus commonly grow on cereal, contaminating the grains with toxins and making them unsafe for consumption [3,18]. Concerning the repercussion on human and animal health, *Fusarium* may disturb the immune system, either by immunotoxic impairment or sensitization with allergy. Aside from the toxic effects, allergic consequences were also reported, such as bronchial asthma, allergic alveolitis and rhinitis, atopic conjunctivitis, organic dust toxic syndrome and chronic fatigue-like syndrome [19]. There are several *Fusarium* allergens, some of them known for cross-reacting to each other [20]. *Fusarium* species are also known for their ability to infect either immunocompetent or immunocompromised individuals [21]. *Fusarium solani*, for instance, contains several allergens that were found to be reactive with serum from patients sensitized to many fungi [22].

*Curvularia* is a relevant genus comprising at least 40 saprophytic species, but only a few of those are known for their capacity to become phytopathogenic. These species produce several mycotoxins with cytotoxic activity as curvulins and brefeldins [3]. Brefeldin A is, in fact, used for that property as a blocker of the intracellular cytokine transport in different immunological studies [23]. Moreover, *Curvularia lunata* was reported to cause eye and skin infections upon trauma [24], as well as onychomycosis, skin ulcerations and subcutaneous mycetoma [25]. *Curvularia* may also sensitize human individuals, causing especially respiratory signs [26] and showing marked cross-reactivity with *Alternaria alternata* and *Epicoccum nigrum* [27,28]. Dogs were also reported to have either infection by *Curvularia* fungi [29] or allergy upon sensitization to their allergens [30]. 

*Cladosporium* is a ubiquitous genus and can be isolated from different materials, such as organic matter, soil, straw, textiles and even ink. It may damage fresh vegetables and fruits, producing great economic losses [3]. Infection by *Cladosporium* fungi has been reported in several species, such as humans, dogs [31], horses [32] and cats [33,34]. Furthermore, allergic conditions associated with *Cladosporium* are currently referred to in humans [3,35] and dogs [36].

*Mucor* and *Rhizopus* are two other genera belonging to the Mucorales group, comprising pathogens of plants [3] that may also affect humans, mainly immunosuppressed individuals [37], as well as allergic individuals, either humans [16] or animals [30,38,39]. 

Another relevant fungal group of diseases is dermatophytosis. This zoonosis is mainly caused by fungi from the genera *Microsporum*, *Trichophyton* and *Epidermophyton* and is rather common among humans and animals attending dermatological consultation, frequently affecting immunocompetent individuals. The prevention and treatment of dermatophytosis rely on good sanitation and hygiene, as well as on specific treatment. It is frequently called the ringworm disease for its round-shaped skin lesions and, besides the etiotropic therapy, vaccination with the first generation of live attenuated preparations have allowed successful control and even eradication when large numbers of cattle and fur-bearing animals were affected [40,41].

The genus *Candida* comprises over 200 species, mostly integrating the normal human and animal microbiota, being considered as commensals but facultatively pathogenic. Only 15 have been isolated from human and animal infections, with *Candida albicans*, *C. glabrata*, *C. parapsilosis*, *C. tropicalis* and *C. krusei* as the most common pathogenic species. Different domestic and sylvatic species (e.g., birds, cats, dogs, horses, pigs and ruminants) have been diagnosed with candidiasis, affecting several organic departments besides mucosa or skin [42]. Unlike malasseziosis, candidiasis is a rare infection in animals but may also occur associated with atopy as well as to other mostly immunosuppressive disorders [43,44]. However, disseminated candidiasis without apparent predisposition has been reported [45].

## 3. Fungi as Sources of Allergens

Allergy to *Dermatophyte* fungi has also been reported. Several allergens from the genus *Trichophyton* have been identified with evidence of *Trichophyton*-related IgE-mediated asthma in humans. In an individual, the same antigens that do not elicit immediate hypersensitivity may nevertheless trigger delayed-type hypersensitivity. Based on the observation of acute vs. chronic skin infection, delayed responses appear to confer protection, while immediate ones do not. Amino acid sequence identity of *Trichophyton* allergens suggests a dual role of these proteins for fungal pathogenesis and allergic etiopathogeny. Some T-cell epitopes have been mainly associated with delayed hypersensitivity, which may be useful for the development of rather effective peptide vaccines, allowing better control of *Trichophyton* infection and related allergy [46].

Regarding sensitization and possible subsequent allergy, fungal spores are between the first substances found as sensitizing to humans, following contact in indoor or outdoor environments. The sensitization to fungal species commonly represents more than 5% of the general population but reaches higher rates in atopic individuals. Exposure to fungal allergens may occur by contacting intact spores and mycelia or their fragments. Spores in germination are known for presenting a wider allergen range. Studies on the genus *Alternaria*, probably the most studied from all allergenic fungi, have been very helpful in terms of the effect of common long-term low-level fungal exposure. Fungal exposure does not mean sensitization or any other pathology, as demonstrated when the exposed population did not include atopic individuals. In fact, indoor fungal exposure and respiratory disease are frequently associated with an atopic predisposition [16] or immune-compromising conditions [3].

Sensitization to *Alternaria* has been estimated to be 7%, while 6% to *Aspergillus* [47], but considering the occurrence of subclinical sensitization, those figures may be underestimated. Furthermore, sensitization to fungi is a considerable cross-reactive condition. Hence, contact with primary sensitization to a limited number of fungal species could result in sensitization to a wide variety of other fungal species, as it was suggested when 6565 individuals with positive IgE in at least one fungal test were tested with a larger battery of fungal species, showing positive to all, in 1208 cases [48]. In fact, fungal proteins sharing homologous structural elements and similar functions showed marked cross-reactivity [49,50]. Fungal structure-derived particles may become aerosolized in concentrations 300–500 times greater than spores [51], which may potentiate contact, leading to possible sensitization. 

Thus far, in Allergome—allergen database (http://www.allergome.org/, accessed on 20 January 2022)—there are 1024 registrations for “fungi” out of 7535 entries. Approximately half of those refer to *Alternaria* with 309 and *Aspergillus* with 195, with respectively 186 and 142 allergens, including isoforms [52]. This makes *Alternaria* the most relevant fungal genus for allergies by far. It is the fungus with the most sensitized humans and the one with the highest association with asthma deaths [53].

Fungi may present the highest concentration of airborne allergen particles, but there is evidence that increased exposure to indoor microbial diversity, including fungi, may represent a protective issue regarding the occurrence of atopy [16], falling into the paradigm of the hygiene hypothesis [54]. Regarding the prevalence of airborne fungi, in northern regions, the amount of fungal spores per cubic meter of outdoor air is usually low during Spring, rising with rainfall and with temperature until a peak during Autumn (around 50,000/cubic meter of air), while in southern regions levels tend to stay more constant, around tens of thousands, varying according to environmental humidity [55]. Regarding the indoor concentration of fungi spores, it usually correlates with outdoor figures, despite major genera, such as *Chaetomium* and *Stachybotrys,* not correlating with outdoor concentrations. Furthermore, major genera associated with the indoor environment, such as *Aspergillus* and *Penicillium*, also do not correlate as much with outdoors as *Cladosporium* and *Alternaria* [56].

Regarding mold, or even house-dust mites and insect allergens for animals, especially dogs and horses, there are not many reports, and major allergens may also differ from those to humans. Despite the evidence of sensitization to mold allergens in dogs, leading to atopic dermatitis, the reported sensitization rate is different between studies, which may be due to a low level of standardization of allergen extracts, resulting in poor specificity of the assays [57].

In equine, recurrent airway obstruction (RAO) has been associated with exposure to moldy hay. Despite sensitization to fungi and aggravation of clinical signs following contact with moldy hay or challenges with mold extracts, only non-IgE-mediated mechanisms have been implicated in the pathogenesis of RAO. However, basophil histamine-releasing test upon stimulation with fungal allergens showed higher in horses with RAO than in healthy individuals [58]. In fact, increased *Aspergillus fumigatus*-specific IgE and IgG responses were found in the bronchoalveolar lavage fluid of RAO-affected horses, following in vitro provocation with fungal extracts [59]. Specific IgE to recombinant allergens, such as Alt a 1 and Asp f 7, 8 and 9, was mostly detected in bronchoalveolar lavage and serum from RAO-affected individuals, despite no difference in specific IgE to fungal extracts between healthy and affected horses. Specific IgG to *Aspergillus fumigatus* has been detected in both healthy and RAO-affected individuals, but the latter were found to have higher IgG levels than Asp f 8 [60,61]. Despite the lack of knowledge about which proteins are major allergens for horses, significant differences in specific IgE against Asp f 7 were found between RAO-affected and healthy individuals. In a study by Scharrenberg et al. (2010) [62], those kinds of differences were only observed in the offspring from one stallion, suggesting a genetic predisposition to sensitization and allergy. In fact, the genetic evaluation identified different quantitative trait loci associated with RAO.

A study Carried out by Leocádio et al. (2019) [63], in a population of 21 horses with a compatible clinical history of allergy, revealed positive intradermal tests (IDT) for *Alternaria alternata*, *aspergillus fumigatus* and a fungi mix in, respectively, nine, five and five individuals. The determination of serum-specific IgE revealed positive to *Aspergillus fumigatus* in four individuals, but no concordance was observed with previous IDT [64].

Relevant mold allergome for dogs or cats has not been clarified yet, and recombinant mold allergens have not been used for diagnostic purposes in these species. For horses, there are already a few mold allergens identified, but a significant rate of recognition to point out the major allergens is still not established [57].

Regarding the nature of immune response against antigenic structures, it is necessary to have in mind that all body epithelial barriers represent ecosystems in which the microbiota (bacteria, fungi and viruses) find nutritive conditions to multiply. These surfaces are consequently highly populated by those producing several metabolites that influence the host immune system, inducing either tolerance or triggering defensive mechanisms as sensitization, sometimes leading to allergy. Healthy immunity relies on a good equilibrium between the microbiota and host defense system, simultaneously preventing invasion by pathogens and avoiding host overreaction. For this purpose, two opposite pathways—immune activation by microbial metabolites and immune regulatory processes—constantly stand in a tiny equilibrium [65]. With regard to animal atopic dermatitis, defects in the lipid and protein constitution of the skin, aggravated by inflammation, may contribute to the impairment of the barrier function, favoring the deep penetration of allergens, with stimulation of the immune response. On the other hand, if a marked genetic predisposition to develop a Th2 kind of immune response is present, even a low epidermal penetration of allergens may trigger sensitization with subsequent allergy. This dual condition is currently designated by the outside/inside–inside/outside paradigm [11]. 

The highest concentration of immune resources is found in the gastrointestinal tract, where a rich mix of commonly commensal bacteria, archaea, fungi and viruses is found. Therefore, its role in the host health/disease equation is crucial but poorly understood [66]. The human gastrointestinal tract is recognized as the first barrier towards food-derived contaminants, including a large variety of xenobiotics. The gastrointestinal tract immune system must face all the related challenges to keep the mucosal barrier up, supporting its structural integrity [67]. Ironically, despite mycotoxin action possibly affecting immune response, *Alternaria alternata* toxins may also contribute to the epithelial barrier function by activating the aryl hydrocarbon receptor pathway [68]. Neonatal gut increase in particular fungi, such as *Candida* and *Rhodotorula*, with a decrease in bacteria, such as *Bifidobacterium*, *Akkermansia* and *Faecalibacterium*, has also been associated with a reduction in T cell expression of Foxp3, CD25 and CD4, leading to an increased risk of childhood atopy [69]. 

Regarding the genus *Candida*, the Allergome platform presents 21 identified allergens out of 7535 entries [52]. Eleven of those are specifically from *C. albicans*, but to the author’s knowledge, no allergens have been associated with possible animal sensitization to *Candida* species.

For diagnostic skin testing, commercial whole-allergen extracts usually vary in the content of major and minor allergens, compromising the reproducibility of the results. Molecular allergens (naturally purified or recombinant) have been produced for nearly all relevant allergen sources, such as pollens, mites, fungi, *Hymenoptera* venom and different foods, and can be used for diagnosis [70]. For veterinary allergy diagnosis, IDT is the first choice complementary method and most suppliers present well-defined concentrations for their allergen extracts. However, standardization is still a relevant issue when aiming for the reproducibility of results [71]. The current, most relevant information regarding sensitization and allergy to fungi in veterinary medicine is summarized in Table 1.

## 4. *Malassezia*, a Complex Big Issue in Animal Allergy

Regarding fungi, the most frequent species found in human and animal skin belong to the *Malassezia* genus, a lipophilic group of yeasts [70] comprising 18 species [73]. Not many phenotyping-based tests are available to identify different *Malassezia* species, frequently allowing several overlaps. The current identification of *Malassezia* yeasts is possible by molecular methods, such as the sequencing of the D1/D2 domain of the large subunit of the rRNA gene, ITS, IGS, CHS2 and β-tubulin genes, allowing the identification of genotypes possibly associated with host-adaptation virulence. Multiplex PCR and MALDI-TOF mass spectrometry are also recognized methods allowing the identification of *Malassezia* species from the skin or in culture, respectively [73]. In the Allergome—allergen database (http://www.allergome.org/, accessed on 20 January 2022 [52]—there are 54 registrations for “*Malassezia*” out of 7535 entries, including 11 species already related to sensitization and 42 allergens, including isoforms. Some of these allergens may be responsible for pro-inflammatory immune response by interacting with dendritic or T cells, probably through Toll-like receptor 2. Specific IgE to *Malassezia sympodialis* allergen Mala s 11, a Mn superoxide dismutase, is correlated with the severity of atopic dermatitis, supposedly by inducing the release of several pro-inflammatory cytokines, such as interleukin (IL)-6, IL-8, IL- 12p70 and TNF-α, by dendritic cells [70]. Mala s 11 is also known for its capacity to activate autoreactive T cells and cross-reactivity with *Aspergillus fumigatus* (Asp f) 6, electing the detection of specific IgE to Asp f 6 as a possible marker for autoreactivity in atopic dermatitis. An allergen (MGL_1304) from *Malassezia globosa* was found to be able to activate mast cells, leading to degranulation and inducing basophils to release IL-4, a trigger interleukin in the pathway to IgE synthesis [61].

In a study by Di Tommaso et al. (2021) [74], all (*n* = 45) dogs subjected to sera determination in an indoor allergen species panel were positive for at least one house-dust or storage mite, 12 for at least one mold species (1 to *Malassezia*), 11 to *Malassezia* and 1 to flea saliva. In fact, *Malassezia* had already been demonstrated to trigger a hypersensitivity response in atopic dogs [71]. As referred by Di Tommaso et al. (2021) [74], the Serum Allergen-Specific IgE Test (SAT) to detect specific IgEs from *Malassezia* revealed a range of positivity between 0% and 60%. In another recent study, evaluating both IDT and SAT with *Malassezia*, there was 24% positivity in IDT, whereas no positivity was found in SAT [79]. Another study reported a percentage of positivity of 35% in IDT [80].

A retrospective study of 111 allergic dogs (60 males and 51 females; 33 from predisposed breeds; 74% indoor and 25.2% outdoor; 59% with mainly seborrheic disruptive skin barrier) living in the inland region of Londrina, Brazil, revealed 49.6% of patients with *Malassezia* overgrowth-associated dermatitis, mostly with atopic dermatitis [76]. A similar study also performed in Brazil, in the São Paulo region, with 84 allergic dogs (45 males and 39 females; 31 from predisposed breeds; 77.4% indoor and 22.6% outdoor; 69% with atopic dermatitis) revealed 58.3% patients with *Malassezia* overgrowth associated to dermatitis [77]. In a veterinary allergy outpatient consultation in Évora, Portugal, of 90 allergic dogs, 14.4% presented positive IDT to *Malassezia*, 6.7% simultaneously to other fungi. In SAT, from 77 allergic dogs, 16.9% presented with specific IgE to *Malassezia*, with 5.2% simultaneously to other fungi [72]. 

*Malassezia pachydermatis* is a commensal inhabitant of canine and feline skin and mucosae, but there are also other species, such as *Malassezia nana* (more associated with outer ear) and *Malassezia slooffiae* (more frequent in the claw fold). Despite this commensal frame not causing lesions, in the presence of different factors, such as the host´s innate and adaptive immune defenses, and expression of the cell wall and secreted virulence factors, *Malassezia* populations may overgrow, causing more or less severe inflammation, in a complex homeostatic equilibrium. This immune-mediated hyperreactivity configures the *Malassezia* dermatitis condition [73], which is rather common in allergic dogs, even without confirmed hypersensitivity to *Malassezia* [78]. 

When that highly demanding homeostatic equilibrium is disrupted, namely with excessive sebum production, diminished sebum quality, moisture accumulation, disrupted epidermal surface or concurrent dermatitis, conditions may become favorable for *Malassezia* overgrowth. Cutaneous inflammation with altered sebum production may frequently occur in conditions, such as skin allergies, for example, atopy, food hypersensitivity and flea allergy; keratinization disorders, with seborrhea and pyoderma; endocrinopathies, such as hyperadrenocorticism, hypothyroidism and diabetes mellitus; metabolic diseases, such as zinc-responsive dermatosis and superficial necrolytic dermatitis; and cutaneous or internal neoplasia, creating the adequate skin microenvironment for *Malassezia pachydermatis* overgrowth [81].

Testing hypersensitivity to *Malassezia* is commonly performed by IDT and serology. However, the lack of standardization of the available extracts makes it somewhat less reliable [73].

There are several dog breeds, namely American Cocker Spaniel, Australian Silky Terrier, Basset Hound, Boxer, Dachshund, English Poodle, Setter, Shih Tzu and West Highland White Terrier, showing increased risk of *Malassezia* overgrowth with consequent dermatitis. Among cats, Devon Rex and Sphynx are also considered for their predisposition. Concomitant diseases, mostly allergic but also other skin or endocrine pathologies, constitute a risk factor for *Malassezia* dermatitis in dogs and cats [69]. *Malassezia* overgrowth is also frequently diagnosed in cats suffering from visceral paraneoplastic syndromes [82].

Clinical dermatitis due to *Malassezia* overgrowth usually presents as pruritic ceruminous otitis externa and kerato-sebaceous scale. The skin develops erythematous lesions, especially on folded areas, which constitute a common risk factor for localized disease, either in dogs or cats, where intertriginous dermatitis will develop [82]. Common clinical signs of chronic dermatitis, associated with *Malassezia* overgrowth in allergic dogs, are presented in Figure 1. Regarding zoonotic risk, it is low, especially in immunocompetent people, as shown by scarce *Malassezia*-derived conditions in humans and by PCR assessment. Good hand hygiene should be considered when contacting dogs and cats presenting *Malassezia* overgrowth [73].

## 5. Main Conclusions

Fungi are ubiquitous forms of life, mostly phytopathogenic and commensal in humans and animals. However, in the presence of impaired conditions, especially concerning skin and mucosal barrier or immune competence, fungi may cause disease by infecting/intoxicating or inducing sensitization with allergy in the presence of a genetic predisposition. The severity of fungal diseases may vary from mild to severely life-threatening. Environmental conditions, such as humidity and temperature, are well-known promoters of fungal development, and the amount and continuity of microbial pressure are also known to play a detrimental role in the effectiveness of animal defenses. Towards allergy to fungi, *Alternaria* and *Aspergillus* are probably the most involved genera of molds for humans and animals. However, in animals, the *Malassazia* genus comprises a group of yeasts with a remarkable responsibility in inflammatory dermatological conditions, mainly in allergic patients, where the skin barrier is found to be frequently impaired. Several direct diagnostic methods are currently used, as well as more state-of-the-art molecular biology ones, allowing more precise species-specific results. Prophylaxis should rely on eviction measures to avoid contact with fungi, in which environmental measures to reduce their population stand very helpful. Attention should be given to the re-establishment of impaired skin and mucosa barrier function, as well as to immune status. 

The modern approach to skin barrier restoration currently focuses on the use of food supplements with nutrients, such as pantothenate, inositol, nicotinamide, choline and histidine (the “PINCH” protocol), which has been shown to increase the cutaneous synthesis of ceramides and decrease transepidermal water loss, even in healthy animals [11]. The administration of topical sphingosine, a pro-ceramide, either in regular baths or in lotions, by promoting the restoration of epidermal lipid lamellae and promoting the reduction of microbial load, controlling superficial infection processes, will result in effective advantages for skin health in dogs [11]. In humans, where the intake of glutathione-GSH-C4 and tocopherol inhibits lipid peroxidation, reducing the oxidative stress associated with inflammatory disorders, topical administration of glutathione-GSH-C4 plus hyaluronic acid proved useful in the presence of chronic seborrheic dermatitis [83]. 

In an experimental study on dog filaggrin and corneodesmosin, two relevant proteins for skin barrier health, carried out by Pin et al. (2019) [84] with healthy Beagles, Golden Retrievers and Labradors, when pro-inflammatory Th2 cytokines were added to cell cultures from the dorsal area and foot-pads, a remarkable decrease in filaggrin expression was observed. It was also found that both the sequence and structure of filaggrin can vary significantly between breeds and even between individuals. This distribution of skin proteins is essentially similar in humans and dogs, suggesting that both species can mutually serve as a skin barrier model. 

Mechanisms involved in sensitization and allergy to yeasts, such as *Candida*, may be rather different from those related to proteinase activity, which is common in molds. In fact, by conducting a study with an experimental mouse model of pulmonary infection with a *Candida albicans* mutant, Wu et al. (2021) [85] observed that in addition to hemostasis, platelets promoted protection against *C. albicans,* following activation by the peptide-toxin candidalysin. Candidalysin-activated platelets would promote Th2 and Th17 cell responses that, despite the reduced lung fungal infection consequences, may predispose to an allergic response. 

Nevertheless, mechanisms associated with the control of fungal microbiota, avoiding infection but not eliminating the fungal population, have been demonstrated regarding *Aspergillus fumigatus*. In fact, Montagnoli et al. (2006) [86], using a mouse model of *A. fumigatus* infection, presented the evidence that fungal antigen presentation by dendritic cells to T CD4 helper cells, under the supervision of T CD25 regulatory cells and involving CD28/B.7-dependent costimulatory mechanisms, would lead to the production of interleukin (IL) 10 and cytotoxic T lymphocyte antigen 4 (CTLA-4), resulting in neutrophil suppression with inflammatory moderation. By stimulating the indoleamine 2,3-dioxygenase (IDO) pathway, the early production of IFN-γ would condition dendritic cells to the further activation of a population of tolerogenic T regulatory cells, which would produce IL-10 and transforming growth factor beta (TGF-β), inhibiting the Th2 pathway and preventing the development of sensitization and allergy to *A. fumigatus*.

Environmental rehabilitation, cleaning and disinfecting surfaces, skin and mucosa antisepsis and topical or systemic antifungal medication are the most useful measures to fight the fungal problem. Specific immunotherapy to fungi is also a developing resource and a possible useful co-measure in fungal disease control.

## Figures and Tables

**Figure 1 jof-08-00235-f001:**
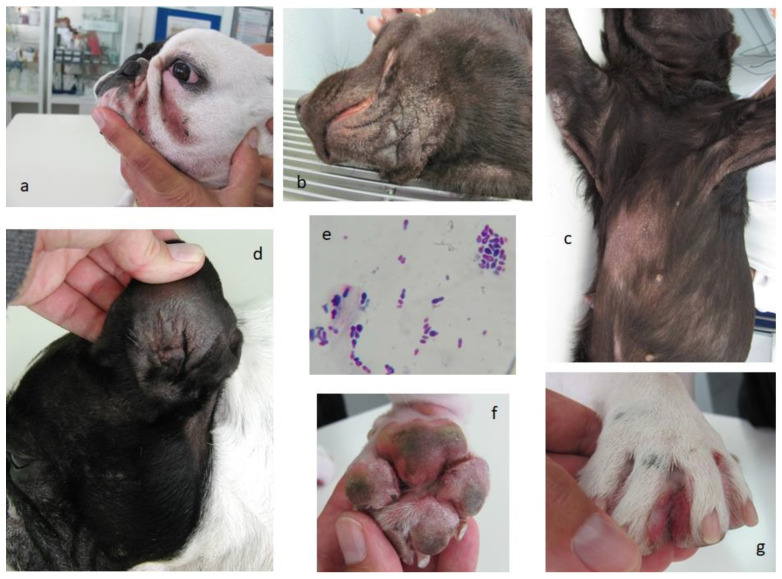
Chronic dermatitis in allergic dogs presenting *Malassezia* overgrowth. (**a**) Facial intertriginous dermatitis; (**b**) Lichenification associated with chronic dermatitis; (**c**) Hyperkeratosis and early lichenification in chronic dermatitis; (**d**) Ear pinnae chronic dermatitis with lichenification; (**e**) Skin cytology showing *Malassezia* overgrowth (400x) (Diff-Quick stain); (**f**,**g**) Chronic interdigital dermatitis.

**Table 1 jof-08-00235-t001:** Summary of the current, most relevant information on the sensitization and allergy to fungi in veterinary medicine.

Sensitizing Species				
	Allergens/Molecular Weight (kDa)	Relevant for	Recommended Extract Concentration/mL	References
*Alternaria alternada*	Alt a 1 (30)	Dog; horse	1000–8000 PNU(#)/100 µg(##)	[59,63,72]
*Aspergillus fumigatus*	Asp f 7 (27.4); Asp f 8 (11); Asp f 9 (34)	Dog; horse	1000–8000 PNU(#)/100 µg(##)	[59,60,61,62,63,64]
*Malassezia* sp.		Dog	100 µg (##)	[72,73,74,75,76,77,78]
*Malassezia globosa*	MGL_1304 (26)	Horse		[61]
*Aspergillus* mix (*)		Dog; horse	1000–8000 PNU(#)/100 µg(##)	[63,72]
Fungi mix (**)		Dog; horse	1000–8000 PNU(#)/100 µg(##)	[63,72]

(*) Extract mix of *Aspergillus flavus*, *A. fumigatus*, *A. nidulans* and *A. niger* (Nextmune, Lelystad, Nederlands). (**) Extract mix of *Alternaria alternata, Aspergillus fumigatus* and *Cladosporium herbarum* (Nextmune, Lelystad, Nederlands); (#) Protein Nitrogen Units; (##) Extracts from Nextmune, Lelystad, Nederlands.

## Data Availability

Not applicable.

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
