# Peer review of "Allergy to Fungi in Veterinary Medicine: Alternaria, Dermatophytes and Malassezia Pay the Bill!"

_jof, 2022, doi:10.3390/jof8030235_

Round 1

Reviewer 1 Report

Dear author,

your manuscript on fungal infections in animals was very interesting and caught my attention.

However, from the reading of the first lines, the text seems to appear more like an editorial than a revision (narrative, systematic?). Moreover, the logical structure of the text may not be suitable for non-expert readers in the field you should divide your manuscript into sections based on fungal pathogens or on human vs non-human and include specific allergic diseases in these sections.

  • Abstract: all the main points are reported in the abstract, but a logical scheme to follow is still missing. it is advisable to better structure the abstract and possibly divide it into sections.
  • If your manuscript is a review, provide a Material and Methods section, with inclusion-exclusion criteria, databased.
  • in the text, there are sections in italics and others in capital letters without rationale. Please double-check the entire front of the item carefully
  • Alternaria spp. is the onychomycosis agent in rare cases. More frequent are Aspergillus spp.-related onychomycosis.
  • In dermatophytes, please cite the Epidermophyton
  • There are no tables or images. Summary tables must be inserted, in which the etiological agents of the pathologies are reported with relative references, or tables reported human and non-human pathogens and related diseases.
  • The introduction is lacking some essential points, please provide a deeper explanation about fungal pathologies. Moreover, at the end of the introduction you must clarify the aim of this manuscript.
  • In section 2, I suggest rewriting and dividing on fungal species, or in human and other animals sections.
  • Line 222-227 lacking in references.
  • Et al. no italic style, please.
  • No data reporting about Candida spp. infections?
  • To improve the introduction and the discussion, read and cite:

- Cosio T, Gaziano R, Zuccari G, Costanza G, Grelli S, Di Francesco P, Bianchi L, Campione E. Retinoids in Fungal Infections: From Bench to Bedside. Pharmaceuticals (Basel). 2021 Sep 24;14(10):962. doi: 10.3390/ph14100962. PMID: 34681186; PMCID: PMC8539705.

- Montagnoli C, Bozza S, Gaziano R, Zelante T, Bonifazi P, Moretti S, Bellocchio S, Pitzurra L, Romani L. Immunity and tolerance to Aspergillus fumigatus. Novartis Found Symp. 2006;279:66-77; discussion 77-9, 216-9. PMID: 17278386.

- Campione E, Mazzilli S, Lanna C, Cosio T, Palumbo V, Cesaroni G, Lozzi F, Diluvio L, Bianchi L. The Effectiveness of a New Topical Formulation Containing GSH-C4 and Hyaluronic Acid in Seborrheic Dermatitis: Preliminary Results of an Exploratory Pilot Study. Clin Cosmet Investig Dermatol. 2019 Dec 16;12:881-885. doi: 10.2147/CCID.S231313. PMID: 31920359; PMCID: PMC6930516.

Reviewer 2 Report

The role of fungal species in development of infections and allergies in Vet sciences is a important area of research addressed by the author. This review provides an overview of the fungal species identified to play a role. The story is interesting but needs extensive English editing and some clarification of issues as pointed out below.

do not use fungi but fungal in: see line 6; 21; 29 2x; 296. 

line 7 commensals

line 7 specify the conditions, the author refers to immunocompromised conditions..

line 13 rephrase sentence

line 18 ...and evolved approximately 1.5 b years

line 21-26 and also 47-50: why is the text in italics?

line 31 remove surrounding, that's obvious from the next word; write into their environment

line 33 for subsequent fungal absorption

line 36 change to ... hyphal fragments

line 36-37  rephrase to for eg reaching almost all places on earth 

line 39 A fumigatus is the most important one and should be added in this list; do the authors mean the most common fungal species in birds? this sentence should be adapted 

line 42 ..from the genus Aspergillus...

line 60  ..disturb the immune...

line 61 you do not say sensitization to allergy...
one becomes sensitive to—and allergic to—a particular substance which is called sensitization

line 64 ...cross-reacting ... with what? the sentence seems not complete, explain this better

line 99 .... that does not..... instead of who do not

line 104 ...which may be useful..

line 105 replace efficacious for effective 

line 112  .. wider allergen frame. this is not clear, is this... wider allergen range.?

line 115-116 ...as it has been .......this part is not clear and should be rephrased in proper English

line 119 ...estimated to be 7%

line 131 ...respect to.. is this refer to...?

line 179 I would not call microbiota xenobiotics, these latter compounds are chemicals and not not find nutritive conditions. If something else is meant this sentence should be changed

line 192 ...other way, other hand?

line 206-208 rephrase to proper English

line 298..a not good role...? what is meant here?

line 303 diagnoses .. diagnostic methods..

line 304 define specific results. typing of species?

line 307 should also be attended? rephrase this, I believe  this is meant: more attention should be given to re-established impaired barrier function..

line 309 remove ..directed ..

reference 6 :wrong title: A case of aspergillosis in a broiler breeder flock
also a mistake in ref 7

ref 22 title is in italisc?

ref 26 mistake in name fungal name

ref 32 name fungal species in italics

Round 2

Reviewer 1 Report

Dear author,

just minor corrections must be addressed. the mauscipt quality has been improved and it is more clear. 

Read and cite:

- Cosio T, Gaziano R, Zuccari G, Costanza G, Grelli S, Di Francesco P, Bianchi L, Campione E. Retinoids in Fungal Infections: From Bench to Bedside. Pharmaceuticals (Basel). 2021 Sep 24;14(10):962. doi: 10.3390/ph14100962. PMID: 34681186; PMCID: PMC8539705.

Author Response

The author thanks the reviewer for the last revision and gives his answer:

The English was once again improved, now as much as possible, avoiding any significant changes after the previous major review.

The author truly does not feel citing the remaining paper (Cosio et al, 2021) as relevant for the frame of the present manuscript. If it results compulsory for the acceptance of the manuscript, the author will do his best to find an acceptable citation, but will feel really uncomfortable. 

Reviewer 2 Report

This paper has been updated very well, the author responded in a correct way the the reviewers remarks

Author Response

The author thanks the reviewer for his/her very useful contribution to the final manuscript.